# Cold Atmospheric Plasma Improves the Colonization of Titanium with Primary Human Osteoblasts: An In Vitro Study

**DOI:** 10.3390/biomedicines12030673

**Published:** 2024-03-18

**Authors:** Madline P. Gund, Jusef Naim, Antje Lehmann, Matthias Hannig, Markus Lange, Axel Schindler, Stefan Rupf

**Affiliations:** 1Clinic of Operative Dentistry, Periodontology and Preventive Dentistry, Saarland University, 66421 Homburg, Germanymatthias.hannig@uks.eu (M.H.); 2Leibniz Institute of Surface Modification (IOM), 04318 Leipzig, Germanyaschindler@t-online.de (A.S.); 3ADMEDES GmbH, 75179 Pforzheim, Germany; 4Synoptic Dentistry, Saarland University, 66421 Homburg, Germanystefan.rupf@uks.eu (S.R.); 5Piloto Consulting Ion Beam and Plasma Technologies, 04668 Grimma, Germany

**Keywords:** biological cell activity, cell attachment, cold atmospheric plasma, primary human osteoblasts

## Abstract

Several studies have shown that cold atmospheric plasma (CAP) treatment can favourably modify titanium surfaces to promote osteoblast colonization. The aim of this study was to investigate the initial attachment of primary human osteoblasts to plasma-treated titanium. Micro-structured titanium discs were treated with cold atmospheric plasma followed by the application of primary human osteoblasts. The microwave plasma source used in this study uses helium as a carrier gas and was developed at the Leibniz Institute for Surface Modification in Leipzig, Germany. Primary human osteoblasts were analyzed by fluorescence and cell biological tests (alkaline phosphatase activity and cell proliferation using WST-1 assay). The tests were performed after 4, 12, and 24 h and showed statistically significant increased levels of cell activity after plasma treatment. The results of this study indicate that plasma treatment improves the initial attachment of primary human osteoblasts to titanium. For the first time, the positive effect of cold atmospheric plasma treatment of micro-structured titanium on the initial colonization with primary human osteoblasts has been demonstrated. Overall, this study demonstrates the excellent biocompatibility of micro-structured titanium. The results of this study support efforts to use cold atmospheric plasmas in implantology, both for preimplantation conditioning and for regeneration of lost attachment due to peri-implantitis.

## 1. Introduction 

Cold atmospheric plasma (CAP) has a broad spectrum of medical applications. In dentistry, it can be used to eliminate biofilms [1,2] or for non-contact surface modification to improve the wettability of dental implants [3]. 

Successful implantation requires osseointegration of the implant. This requires the differentiation of progenitor cells into osteoblasts which later become osteocytes embedded in the mineralized matrix. During osseointegration, osteoprogenitor cells and vessels grow and osteoblasts adhere to the implant surface [4]. Therefore, the ability to promote the rapid and effective attachment of the osteoblasts is crucial.

The long-term clinical success rate of titanium dental implants is reported to be 87.8% with follow-up periods of up to 36 years. Successful osseointegration and long-term stability of the implant–bone interface enable such results to be achieved in clinical follow-up assessments [5]. The peri-implant tissue acts as a barrier and protects against oral bacteria. Contamination of the implant surface by microorganisms can be prevented, thus avoiding inflammation of the surrounding tissue and bone [6,7].

There are multiple options to enhance cell proliferation and ultimately osseointegration, including etching, coating, and sandblasting [8,9,10]. Several coating options have been described in the literature. There are nanoparticle coatings with osteo-integrative activity, such as Al_2_O_3_ nanoparticles, hydroxyapatite, and calcium phosphate, nanoparticle coatings with osteo-integrative and antibacterial activities, such as TiO_2_ nanoparticles and nano-crystalline diamond, and nanoparticles with antimicrobial activity such as dental coating materials such as Ag nanoparticles, ZnO nanoparticles, CuO nanoparticle, Quercitrin and Chlorhexidine [11].

Previous research has demonstrated the effects of cold atmospheric plasma pre-treatment of titanium surfaces on the initial attachment of primary human fibroblasts [6]. Over the years, several studies have focused on the effect of treating different titanium surfaces with different plasma jets on osteoblasts [12]. 

Both fibroblasts and osteoblasts show improved spreading on titanium surfaces when treated with cold atmospheric plasma. Faster proliferation and adhesion of immortalized human gingival fibroblasts (HGF-1) and osteoblast-like cells (MG-63) on coated and uncoated titanium and zirconia discs were observed after treatment [13].

However, studies investigating the impact on the initial attachment of primary human osteoblasts are currently lacking. Therefore, the aim of this study was to investigate the attachment and biological activity of primary human osteoblasts on micro-structured titanium surfaces after treatment with cold atmospheric plasma during the first 24 h.

## 2. Materials and Methods

### 2.1. Setting

#### 2.1.1. Titanium Discs

The test specimens consisted of 144 titanium discs (titanium grade 2, Friadent, Mannheim, Germany) with a diameter of 5 mm and a height of 1 mm. The surfaces of the discs were micro-structured through sandblasting and etching to a surface roughness value (Ra) of 2 µm. Of these, 54 discs were left untreated as controls.

#### 2.1.2. Plasma Device

This study used a plasma source developed at the Leibniz Institute for Surface Modification located in Leipzig, Germany. The miniaturized plasma source equipped with a coaxial electrode system was microwave-excited and operated with helium gas. The inner electrode was formed by an inner conductor consisting of a 0.3 mm thick steel tube. The second electrode, which was made of 3.4 mm thick steel and served to shield the generated microwaves, was formed by the outer body of the plasma source. The process gases were added through the inner conductor, and their flow rate was regulated by a mass flow controller. To activate the plasma source, a pulsed microwave generator was used generating at a frequency of 2.45 GHz. The generator allowed for a peak power range of 100–300 W, an average power range of 1–9 W, and a pulse width of 1–10 μs. 

The plasma treatment settings, including the pulse width, pulse power, and average microwave power, could be adjusted as desired. The plasma source was mounted on a computer-controlled 3-axis movement system (Steinmeyer MC-G047, Feinmess Dresden GmbH, Dresden, Germany) to ensure uniformity of treatment. The parameters used for processing comprised a peak power of 300 W and a pulse width of 5 µs, resulting in a mean power of 5 W and a scanning speed of 8 mm/s. The plasma jet was used to process the surface in a meandering line scan. The working distance was set at 2 mm, and a gas flow of 2 L/min was used. Helium was selected as the carrier gas and was mixed with 5 sccm of oxygen. These parameters limited the surface temperature of the titanium to about 40 °C during the plasma treatment. 

#### 2.1.3. Primary Human Osteoblasts

Primary human osteoblasts from femoral trabecular bone tissue from the knee or hip joint region were obtained from PromoCell (Heidelberg, Germany). Cryopreserved cells were cultured in DMEM/10% FBS in culture flasks, with third passage cells being used in the experiments.

#### 2.1.4. Sample Treatment

The samples were treated in a controlled environment in an S2 laboratory where the air temperature was 22 ± 2 °C and the air was changed eight times per hour at a slight positive pressure. After the treatment, primary human osteoblasts were immediately applied to a well plate under a sterile bench. Both the treated samples and the untreated controls received 10^4^ cells. The titanium specimens were kept moist with DMEM-liquid medium for 30 min before the wells were filled with 200 µL of the same medium. The samples were then incubated in an incubator (Hera Cell 240, Thermo Scientific, Waltham, MA, USA) at 37 °C in a 5% CO_2_ atmosphere for 4, 12, and 24 h.

### 2.2. Sample Analysis 

#### 2.2.1. Surface Coverage, Images

Fluorescence microscopy was used to visualize the colonization of the primary human osteoblasts on the titanium discs. Staining was performed with Vinculin, Phalloidin, and DAPI. The cells were fixed to the specimens with a 2.5% glutaraldehyde solution (Carl Roth GmbH, Karlsruhe, Germany) for 30 min. After the glutaraldehyde treatment, the specimens were washed five times with phosphate buffered saline (PBS) to remove glutaraldehyde. Permeabilization of the cell wall was initiated by exposure to 0.1% Triton-X (Roche, Grenzach-Wyhlen, Germany) in PBS. A double rinse step was performed using PBS, and the non-specific background was minimized by using 1% bovine serum albumin (BSA in PBS, Gibco, Invitrogen, Carlsbad, CA, USA). The samples were placed in a humidified chamber and treated with a primary antibody directed against vinculin (Anti-Vinculin, mouse monoclonal antibodies; Sigma-Aldrich Chemie GmbH, Steinheim, Germany) diluted in 5% bovine serum albumin. The samples were then washed twice with 0.1% Tween 20 (Roche, Grenzach-Wyhlen, Germany) in phosphate-buffered saline and the Tween 20 was removed with phosphate-buffered saline. The samples were then treated with a secondary antibody (anti-mouse IgG, RD Systems, Minneapolis, MN, USA) for 1 h while maintaining the humidity in the chamber. This was followed by another rinse with 0.1% Tween 20 and PBS. As a final step the samples were immersed in a phalloidin stock solution (1 mg/mL MetOH) for 25 min at room temperature, in the dark, followed by the application of DAPI solution (Roche, Grenzach-Wyhlen, Germany) for 10 min, also in the dark.

The titanium discs were fixed onto Superfrost slides (Menzel-glasses, Braunschweig, Germany) before being thoroughly examined using a reflected-light microscope (Axio Scope A1, Carl Zeiss, Göttingen, Germany) with 2.5×, 10×, and 20× magnification objectives. A general view was then taken for each sample using a 25× magnification (2.5 (lens) × 10 (ocular)). In addition, high magnification images were taken to provide representative illustrations.

#### 2.2.2. Alkaline Phosphatase Activity

Colorimetric determination of alkaline phosphatase activity was performed using a Biovision kit obtained from Mountain View, California, USA. After extracting 80 µL of medium from each well of culture plates containing titanium discs, the medium was transferred to a 96-well plate manufactured by Greiner bio-one in Frickenhausen, Germany. This was followed by the addition of 50 µL of para-nitro-phenyl-phosphate (pNPP) solution, and the plate was left in the dark at an ambient temperature for 60 min. At the same time, a blank color sample was prepared. The standard solution consisted of 40 µL of 5 mM pNPP solution in 160 µL of assay buffer. A range of increasing concentrations was used to construct the standard curve. The experiments were performed in duplicate. After a one-hour reaction time, 20 µL of stop solution was added, and the absorbance was measured at a wavelength of 405 nm using an ELISA reader (Tecan Infinite 200, Magellan V6.6, Tecan, Groedig, Austria).

#### 2.2.3. Cell Proliferation/Cell Viability Assay

After incubation for 4, 12, and 24 h, each well of the culture plates was treated with 20 µL of WST-1 reagent (Cell Proliferation Kit WST-1, Water-Soluble Tetrazolium). In addition, a control well was prepared containing only medium, without WST-1 reagent. A blank was also prepared containing only the WST-1 reagent. After two hours of incubation under the above culture conditions, the well plate was vortexed for one minute. Light absorption was measured at a wavelength of 420 nm using an ELISA reader, and a reference value was determined at a wavelength of 600 nm.

### 2.3. Statistics

The Mann–Whitney *U* test was performed to compare control and test samples statistically. Statistical significance was considered when the *p*-value was *p* < 0.05.

## 3. Results

### 3.1. Fluorescence Microscopy

Fluorescence microscopy conducted after 4, 12, and 24 h demonstrated the attachment of the primary human osteoblasts on the titanium surfaces. After 4 h of cultivation, the primary human osteoblasts present on the micro-structured titanium surface within the test and control groups adhered to the titanium surfaces with comparable areas of focal contacts (Figure 1A,B). After 12 h, however, differences between the test and control groups were visible. Osteoblasts covered the entire surface of the micro-structured titanium discs treated with cold atmospheric plasma. They were dispersed from the application site, whereas cells on untreated titanium remained in the area of application without extensive spreading (Figure 2A,B). However, the attachment areas of the individual cells on the treated and untreated titanium were comparable. After 24 h, there were no differences in the pattern of cell distribution on the micro-structured titanium surfaces of both the test and control groups. In addition, focal attachment areas remained comparable (Figure 3A,B).

### 3.2. Cell Activity

Statistically significant differences in the alkaline phosphatase activity were observed between the test and control groups at 4 and 12 h (4 h: *p* = 0.013, 12 h: *p*= 0.015), although the differences in the mean values and standard deviations between the two groups were small. No statistically significant differences were observed between the plasma-treated and control samples after 24 h. After 4 h alkaline phosphatase activity was 8.1 (±0.2) U/mL in the control group and 8.5 (±0.4) U/mL in the test group, after 12 h it was 17.4 U/mL (±1.2) in the control group and 19.2 (±1.9) U/mL in the test group, and after 24 h it was 37.2 U/mL (±1.2) in the control group and 36.4 (±4.5) U/mL in the test group (Figure 4).

When the cell proliferation was compared using the WST-1 assay, statistically significant differences were observed between the test and control groups at 4, 12, and 24 h (4 h: *p* = 0.022, 12 h: *p*= 0.036, 24 h: *p*= 0.006). In particular, the OD value after 4 h in the test group of 0.21 (±0.07) compared to 0.11 (±0.01) in the control group showed a clear difference (*p* = 0.022). After 12 h of cultivation, the OD value was 0.31 (±0.07) in the test group and 0.24 (±0.05) in the control group, a difference that was also statistically significant (*p* = 0.036). Even after culturing the osteoblasts for 24 h, there were still differences in cell proliferation with values of 0.32 (±0.03) for the test group and 0.26 (±0.03) for the control group. This difference was also statistically significant with a *p*-value of 0.006 (Figure 5).

## 4. Discussion

Previous studies have demonstrated the favorable response of osteoblasts to biofilm-covered titanium treated with cold atmospheric plasma and mechanical cleaning [1]. Plasma-treated titanium surfaces showed reduced contact angles, larger cell sizes, and better spreading of osteoblastic cells compared to non-treated controls [3]. The beneficial effects of cold atmospheric plasma treatment on titanium surfaces and osteoblasts have been demonstrated on smooth, moderately rough, and rough implant surfaces [14]. In addition, studies have shown the potential to remove biofilms from micro-structured surfaces of titanium implants [15]. In addition, cold atmospheric plasma facilitates osteoblastic cell differentiation, which could serve as an exciting tool for bone regeneration in the future [16]. 

The effects have previously been studied on primary human gingival fibroblasts and have shown that pre-treatment of titanium surfaces leads to improved initial adhesion of these cells [6]. To the best of our knowledge, this is the first study to demonstrate beneficial effects of cold atmospheric plasma to condition micro-structured titanium on the early attachment of primary human osteoblasts.

After 4 h, the cells of both groups showed a comparable attachment area on the treated and untreated titanium surfaces, with cell proliferation being higher in the test group after 4 h, 12 h, and 24 h, as measured by the WST-1 test. However, after 12 h, the cells on the cold atmospheric plasma-treated titanium surfaces had covered the entire surface of the titanium disc, whereas the cells on untreated titanium remained in the application area. However, the attachment areas of individual cells were not visually different in the fluorescence microscopic images. After 24 h, the distribution of cells on both titanium surfaces and their respective attachment areas were identical. Again however, the WST-1 kit indicated higher cell proliferation on the plasma-treated titanium surfaces. The adhesion of human osteoblast-like cells (MG-63) was not altered on titanium grade IV implant surfaces that had been treated cold atmospheric plasma treatment within an observation period of 24 h [13]. 

On air plasma pre-treated titanium MC3T3-E1 cells covered a larger surface area after 2 h. In addition, better cell proliferation and migration were observed with a more developed cellular network. The authors point out that the effects of plasma on cells cannot be maintained for a long time [17]. The results are consistent with ours. It may be necessary to treat implants with cold atmospheric plasma immediately before implantation to improve the success rate [17]. 

Oxygen-plasma treatment of titanium surfaces had no effect on the metabolic activity and proliferation of primary human alveolar bone osteoblasts from day 1 to day 7. Plasma pre-treated surfaces showed comparable cell densities to controls at day 1, 3, and 7. No differences in the organization of actin and vinculin were observed between treated and untreated surfaces [18]. However, Henningsen et al. also describe a higher initial cell attachment (argon plasma) after 2 h in comparison to non-plasma-treated titanium discs. They used murine osteoblast-like cells MC3T3-E1 in vitro. They describe better results (number of cells) for plasma-treated surfaces (argon and O_2_ plasma) compared to controls during a 72 h incubation. They also found better cell proliferation after 24 h, which is in line with our results [19]. Other studies have also confirmed these results [20]. Larger average cell areas (human osteoblasts, MG-63) were described 24 h after the argon-plasma treatment of titanium surfaces. Cells on these surfaces were enlarged, and cells on the untreated surfaces were less adapted. In summary, higher levels of cell proliferation and adhesion were described [21]; the penultimate of which is also consistent with our findings.

Titanium surfaces treated with argon-plasma showed improved adhesion of rat bone marrow cells after 6 h [22]. Lee et al. describe higher number of attached cells (human osteosarcoma cell line) on plasma-treated titanium using a dielectric barrier discharge (DBD) plasma after 2 h. After 5 days the number of cells in the test group was 40.2% higher than in the control group. After 24 h, cells in the test group were more uniformly attached to the implant surface and cell density was much higher [23]. These results are confirmed by Long et al. who found that after 24 h up to 30% more cells (MC3T3-E1 mouse pre-osteoblasts) adhered to the plasma-treated surface [24]. A significant increase in the number of osteoblasts (MC3T3-E1 and MG-63) adhering to argon-plasma-treated titanium surfaces is also described for 10 min [14]. After 12 h, Wang et al. described excellent adhesion and elongation of osteoblast rat cells. The cell area increased after titanium plasma treatment, even after 24 h [25]. Better adhesion of rat bone marrow cells for plasma-treated titanium disks at 1, 3, 6, and 24 h was demonstrated by Ujino [26]. These results are confirmed by Hayashi who investigated the effect of plasma treatment on titanium surfaces and rat bone marrow cells [27]. These results contradict our own. There may be several reasons for the conflicting results described. Firstly, the noble gas and the exposure time of the beam were different. The exposure time plays a crucial role. A higher cell density is described for plasma-treated surfaces after 1 min of surface treatment compared to 12 and 16 min [28]. These results are confirmed by Swart et al. The highest cell adhesion was observed on surfaces that had been treated with argon plasma for one minute. The presence of inorganic contaminants was observed with longer treatment times. This may have influenced the behavior of the cells [29]. Other exposure times have also been investigated in the literature [30]. Han et al. describe a weak proportionality of the number of osteoblasts to the treatment time for the same culture period. A significant increase in the number of cells was observed after 4 min of plasma treatment [20].

Secondly, the surfaces were different. It has been described several times in the literature that bone-related cells prefer rough surfaces. In this case, better initial cell proliferation and adhesion have been demonstrated [31]. The use of rough, micro-structured surfaces in this study may explain why the adhesion of individual osteoblasts was not really improved after plasma treatment compared to the control group. Thirdly, different types of studies were carried out. The study presented was an in vitro study. However, animal studies were also included in the discussion. Also, different cells were used. In this study, primary human osteoblasts were used. The use of primary human osteoblasts provides a more realistic representation than immortalized cells. These cannot replace primary human cells [32]. Similarities in mineralization and cell proliferation between primary human osteoblasts and MC3T3-E1 cells are reported. Furthermore, SaOs2 and MG-63 cells demonstrated a higher proliferation rate than primary human osteoblasts. In addition, SaOs2, but not MG-63 cells demonstrated similar mineralization potential, gene regulation and alkaline phosphatase activity compared to primary human osteoblasts [32].

For alkaline phosphatase activity, the absolute mean values differed only slightly between the test and control groups. Although the values for the plasma-treated titanium samples were statistically significantly higher (at 4 and 12 h), the difference in the mean values was small. The statistical significance here is probably due to the low standard deviations. This indicates well-controlled and reproducible conditions but should not be overinterpreted. The values for the test and control groups after 4 h, 12 h, and 24 h indicate that the primary human osteoblasts accepted the micro-structured titanium as a surface for settling. The alkaline phosphatase activity of rat bone marrow cells was higher at 7 and 14 days in the plasma-treated titanium group than in the control group. Plasma treatment therefore increases alkaline phosphatase activity [26]. These results are also confirmed by Hayashi using rat bone marrow cells, with the best achieved results being obtained using the argon plasma jet [27]. Lee et al. also investigated the alkaline phosphatase activity of the human osteosarcoma cell line used. After 7 days, it was 81.5 % higher than the control [23]. 

To the best of the authors’ knowledge, there is currently no study that has also examined the alkaline phosphatase activity after 4, 12, and 24 h of primary human osteoblasts. A direct comparison with the results in the literature is therefore difficult. In principle, the results obtained so far indicate that the alkaline phosphatase activity increases after plasma treatment of the surface. This is supported by the fact that the literature also describes an increase in cell proliferation compared to the controls. The increase in cell proliferation is consistent with our findings. In this study, cell proliferation did not correspond to the alkaline phosphatase activity, which is a marker of osteoblast activity, early bone differentiation, and bone formation [26]. It is possible that the alkaline phosphatase is comparable because the cells find favorable conditions on both the plasma-treated and the untreated titanium surfaces. The different cell proliferation values indicate that the cells spread more actively on the plasma-treated titanium.

The titanium test specimens were fabricated from pure, medical-grade titanium (titanium grade 2) and had surface configurations typical of intra-osseous implants. Treatment was performed using a miniaturized plasma source that has been previously used and investigated [26]. The experiments were performed under standard environmental conditions. Parameters were carefully selected to ensure that biologically acceptable temperatures were maintained in the treatment area. This allowed for the most realistic possible approximation of real conditions in practice. The cultivation time was limited to a maximum of 24 h in order to avoid the need for a prophylactic antibiotic addition to the medium. The analytical methods used focused on visualizing the attachment, morphology, and biological activity of primary human osteoblasts. Both plasma-treated and untreated control samples showed cell proliferation for up to 24 h. This supports previous research indicating the exceptional biocompatibility of titanium [33].

However, there are clear limitations to this study. This is an in vitro study, so it is not possible to assess the extent to which the effects described might also be observed in a clinical setting (e.g., influence of saliva and blood). The cells were derived from the trabecular bone, not from the mandible, and were therefore purchased rather than harvested from the patient. Studies investigating the effect of primary human osteoblasts of the mandible are needed. The results also have limitations: after 24 h, the beneficial effect of the plasma pre-treatment of titanium on the fibroblasts (cell distribution and alkaline phosphatase activity) appears to be minimal compared to the control. It would be interesting to see if similar results could be demonstrated in vivo. If the results are similar, in the overall assessment, no decisive effect of the plasma pre-treatment of titanium on primary human osteoblasts can be determined. Then this would only be the case initially.

Nevertheless, the results highlight the wide range of potential applications for cold atmospheric plasma in a variety of dental fields. The results of this study may be of value in dental practice. The application of cold atmospheric plasma to titanium increases the wettability of the surface and thus improves the distribution of biological substances and cells applied to the surface. These results are consistent with our own and with other researchers’ findings.

The present in vitro study shows that primary human osteoblasts on micro-structured titanium exhibit a faster cell proliferation and initial homogeneous distribution as well as a higher alkaline phosphatase activity after 4 and 12 h, which opens up the application of cold atmospheric plasmas in implant dentistry. The technique can be used in pre-implantation conditioning and to regenerate lost attachment caused by peri-implantitis [17,34]. Further studies are needed to evaluate the potential and opportunities of cold atmospheric plasma in dentistry, especially in a clinical setting [34]. However, it is undeniable that its implementation offers exciting possibilities, particularly for improving the success of implants. 

## 5. Conclusions 

The results of the in vitro study show that cold atmospheric plasma treatment improves the initial attachment of primary human osteoblasts. After 12 h, osteoblasts had covered the entire surface of the pre-treated titanium, whereas cells on the untreated titanium remained in the area of application. However, the areas of attachment of individual cells were not visually different in the fluorescence microscopic images. After 24 h, the distribution of cells on both titanium surfaces and their respective attachment areas were identical. After 4 h of culture, slight advantages were observed for alkaline phosphatase activity and clear advantages for cell proliferation as assessed by the cell-proliferation WST-1-assay. After a longer culture period of 24 h, the benefits of alkaline phosphatase activity diminished, demonstrating the exceptional biocompatibility of the micro-structured titanium used. In terms of cell proliferation, the experimental group showed improved performance, which was also statistically significant. These results suggest that cold atmospheric plasma may have clinical applications in peri-implantitis treatment and in pre-implantation. Future studies in a clinical setting are required, as the results are not immediately applicable to practice.

## Figures and Tables

**Figure 1 biomedicines-12-00673-f001:**
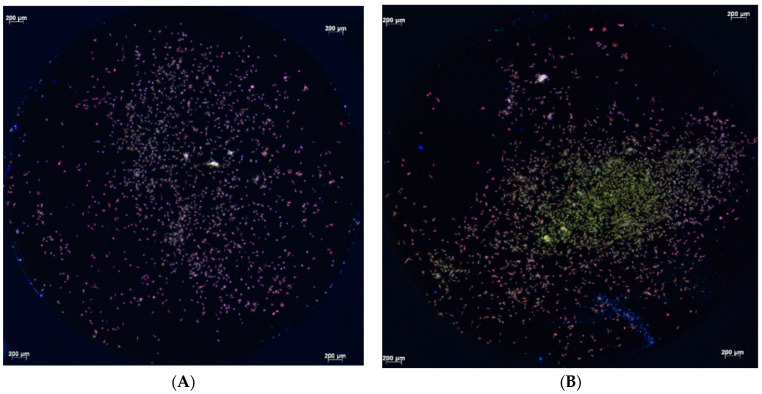
Fluorescence micrographs of primary human osteoblasts on micro-structured titanium discs (**A**) and controls (**B**) after 4 h of cultivation (vinculin, phalloidin, and DAPI staining). The attachment areas of individual cells were comparable (**A**,**B**).

**Figure 2 biomedicines-12-00673-f002:**
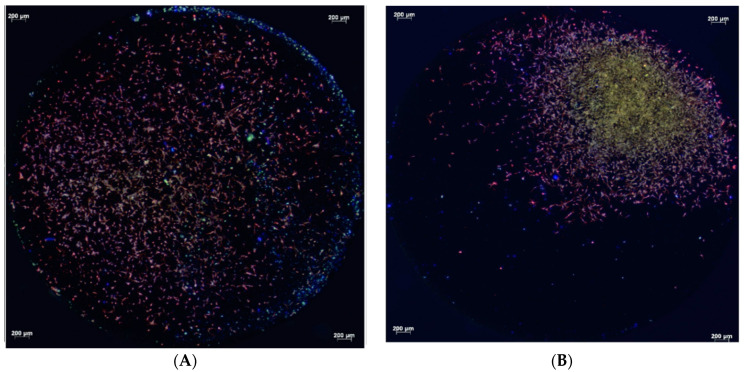
Fluorescence micrographs of primary human osteoblasts on micro-structured titanium discs treated with cold atmospheric plasma (**A**) and control samples (**B**) after 12 h of cultivation (Vinculin, Phalloidin, DAPI stains). The cells are spread homogeneously on the treated titanium surfaces (**A**), whereas the cells on the untreated titanium remained around the application area (**B**). The attachment pattern of the individual cells is not different (**A**,**B**).

**Figure 3 biomedicines-12-00673-f003:**
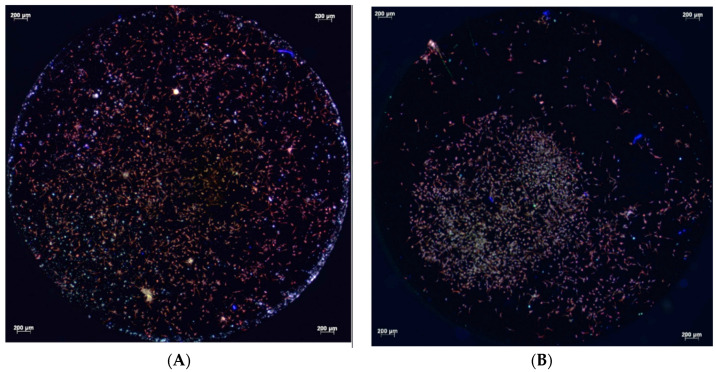
Fluorescent micrographs of primary human osteoblasts on cold atmospheric plasma-treated (**A**) and untreated (**B**) micro-structured titanium discs after 24 h cultivation (vinculin, phalloidin, and DAPI staining). The distribution of cells on the titanium surfaces and their adherence areas show no apparent differences.

**Figure 4 biomedicines-12-00673-f004:**
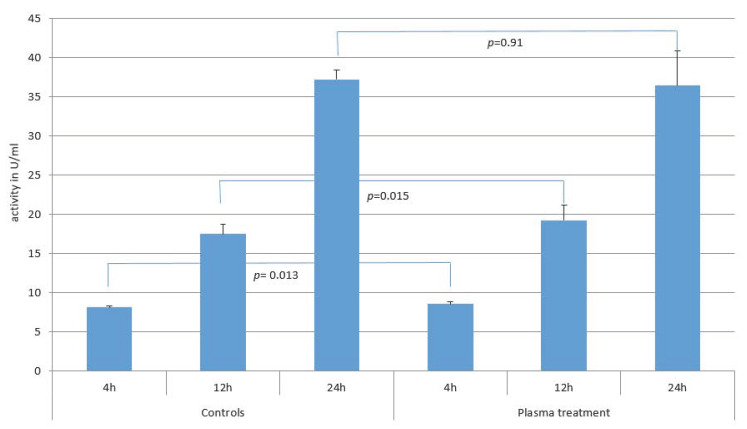
Comparison of alkaline phosphatase activity after 4, 12, and 24 h between micro-structured titanium (plasma treated) samples and controls. The bars correspond to the mean value and the line on top to the ±standard deviation. Statistically significant differences were found for the 4 h and 12 h results (4 h: *p* = 0.013, 12 h: *p* = 0.015). No statistically significant differences were found for the 24 h results (*p* = 0.91).

**Figure 5 biomedicines-12-00673-f005:**
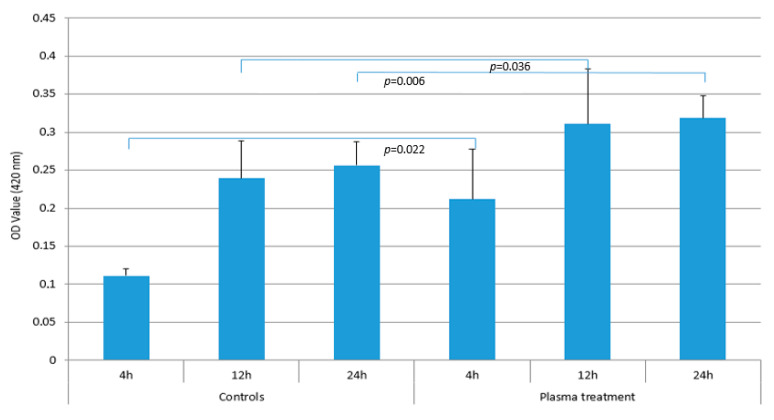
Comparison of cell proliferation using WST-1 assay after 4, 12, and 24 h between plasma treated samples and untreated controls. The bars correspond to the mean value and the line on top to the ±standard deviation. Statistically significant differences were found for the 4 h, 12 h, and 24 h results (4 h: *p* = 0.022, 12 h: *p* = 0.036, 24 h: *p* = 0.006).

## Data Availability

The data supporting the findings of this study are available from the corresponding author upon reasonable request.

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
