# Peer review of "Cold Atmospheric Plasma Improves the Colonization of Titanium with Primary Human Osteoblasts: An In Vitro Study"

_biomedicines, 2024, doi:10.3390/biomedicines12030673_

Round 1

Reviewer 1 Report

Comments and Suggestions for Authors

I have reviewed a manuscript that investigates the effects of cold atmospheric plasma treatment on microstructured titanium surfaces and the attachment of primary human osteoblasts. The study addresses a relevant topic and employs appropriate experimental methods. However, before the manuscript can be considered for publication, there are some issues that need to be addressed.

Major Revisions Needed:

Introduction:

The introduction provides a good background on the potential applications of cold atmospheric plasma in dentistry. Still, more context is needed on the effects of osteoblast attachment to establish the rationale and importance of the current study.

Methods:

More details on the plasma treatment parameters and conditions are needed, such as gas composition and flow rates, treatment times, temperature monitoring, etc. It is also necessary to explain how treatment uniformity was ensured across samples. Additionally, more information is needed on the osteoblast isolation and culture methods, such as the donor source and how cells were maintained and seeded onto the titanium discs. Furthermore, statistical analysis details are very sparse, and it is unclear what statistical tests were used for the different assays and what constituted statistical significance.

Results:

The fluorescence images provided are small and difficult to interpret. Higher-resolution images should be provided. For quantitative assays, absolute values should be provided in graphs or tables, not just reported in the text. Error bars indicating variability should be added. Furthermore, more explanation is needed for the seemingly contradictory results of alkaline phosphatase and cell proliferation at 24 hrs.

Discussion:

The discussion is underdeveloped. The study findings should be placed better into the context of existing literature. Limitations of the study and alternative interpretations of the results should be discussed.

References:

There is heavy reliance on citing internal works. More references to related external studies are needed to support statements in the introduction and discussion. Reference formatting is inconsistent both in-text and in the reference list.

Minor Revisions:

English language usage is generally good, but minor grammatical/word choice issues exist. Figure captions could be improved to convey essential information without referring to the main text. Conclusions could better summarize specific study findings.

Comments on the Quality of English Language

Minor mistakes

Author Response

Attached you will find the rebuttal letter

Reviewer 2 Report

Comments and Suggestions for Authors

The manuscript entitled “Cold atmospheric plasma improves the colonization of titanium with primary human osteoblasts in-vitro” constitutes a highly interesting study. The manuscript is well written, and fits within the scope of the journal Biomedicines (ISSN 2073-4360), in particular to the special issue “Plasma Applications in Biomedicine”. However, the manuscript presents several relevant issues that need to be addressed before its consideration for publication. Additional experiments must also be performed to improve the quality and relevance of the study. With this lower number of experiments, these data can be published as short communication.

Comments:

1.    In the Introduction section, the authors should also include some examples of coating materials used to improve the biological performance of titanium.

2.    The section 2. Materials and Methods must be reorganized in specific subsections.

3.    (lines 109, 110): The formatting must be improved.

4.    (Materials and Methods section, line 139): The authors need to describe the protocol used to extract the ALP from the cells.

5.    Gross and TEM images of the titanium discs must be provided.

6.    The quality of the images in Figures 1, 2 and 3 need to be improved.

7.    In Figure 1 and Figure 3, lower magnification images also need to be provided (as in Figure 2) to support the authors’ claims about the spreading of the cells.

8.    Figures 4 and 5: The quality of the graphs must be considerably improved. Also, the statistical differences described in the figures caption should be clearly identified in the graph.

9.    Longer cell culture experiments of the cells growing on the titanium disks should also be performed to further study the capacity of the materials to improve the proliferation and maintenance of primary human osteoblasts.

10. In the text segments of lines (211-214 and 217-222), the values must be presented with the standard deviation.

11. (Discussion section, line 278): “... activity of fibroblasts”. (???) The authors should revise this.

12. The Discussion section lacks comparisons with other studies from the literature.

13. The improvements of the titanium disks in ALP activity and cell viability when compared to the control samples are minimal. These issues need to be discussed in much more depth in the Discussion section.  

Comments on the Quality of English Language

Minor editing of English language required

Author Response

(The authors gave the same response as above.)

Reviewer 3 Report

Comments and Suggestions for Authors

Dear authors,

Dear authors,

This is a well-conducted original study comprising numerous in vitro experiments, but a few concerns related to the work do exist and are listed below.

 Title

Cold atmospheric plasma improves the colonization of titanium with primary human osteoblasts in-vitro

Consider replace and improve the title should, for example:

Cold atmospheric plasma improves the colonization of titanium with primary human osteoblasts: an in vitro study

1.     Introduction

Line 46. After being treated with CAP, the surfaces' wettability significantly increases.

What is the reference to support this statement.

Line 66: Previous research has demonstrated the effects of cold atmospheric plasma modification on primary human fibroblasts.

This sentence should be improved.

2.     Materials and Methods

The structure of the text must be uniform.

5. Conclusions

What are the limitations of this in vitro study?

It is possible the translation to in vivo studies?

Author Response

(The authors gave the same response as above.)

Round 2

Reviewer 1 Report

Comments and Suggestions for Authors

Overall Evaluation

The authors have addressed many of the major issues raised in my previous review by adding more context and rationale in the introduction, providing more methodological details, improving data presentation, expanding the discussion, and addressing language/formatting concerns. The manuscript quality has improved, and I appreciate the significant effort to revise it accordingly based on feedback.

While still needing some modifications, I now believe this manuscript could be acceptable for publication after major revision instead of my initial recommendation for rejection.

Remaining Major Revisions:

Methods

More specific details are still needed on the plasma treatment parameters - what composition of gas blend was used? How was temperature monitoring done? Provide actual treatment exposure time.

Results

Include actual p-values for alkaline phosphatase & cell proliferation data; don't just state "statistically significant difference."

Add scales to fluorescence microscope images to aid in the interpretation

Explain why alkaline phosphatase vs cell proliferation results seem contradictory

Discussion

Contextualization is still quite limited - relate back to how findings fit with specific previous studies on plasma effects on osteoblasts

Address potential differences due to cell type variations (prior use of immortalized vs. primary cells)

Minor Revisions:

Some language polishing is still needed - watchword choice/awkward phrases

Carefully proofread references - some inconsistencies remain

Conclusions could highlight implications/applications more strongly

In summary, the authors have been responsive to feedback and addressed several of the deficiencies in this study. With some remaining revisions to methods/results reporting and further developing the discussion, I believe this could make a worthwhile contribution to the literature.

Comments on the Quality of English Language

Minor editing of English language required

Reviewer 2 Report

Comments and Suggestions for Authors

The authors have improved the manuscript considerably during this round of revisions. However, the organization of the Materials and Methods section can still be improved with numbered subsections and the quality of the fluorescence microscopic images/graphs provided by the authors is still low and should be enhanced.

Comments on the Quality of English Language

none

Author Response

Please see the rebuttal letter attached.

Round 3

Reviewer 1 Report

Comments and Suggestions for Authors

The authors have done an excellent job addressing the remaining issues in my prior review. The manuscript has improved substantially through additional details on methods and results, as well as a more thorough discussion and conclusion. I believe the study now provides a worthwhile contribution to understanding early osteoblast interactions with cold atmospheric plasma-treated titanium surfaces.

Comments on the Quality of English Language

 Minor editing of English language required

Author Response

Please see the rebuttal letter

Reviewer 2 Report

Comments and Suggestions for Authors

The quality of the images and graphs might still be improved.

Higher magnification fluorescence microscopy images can be added as supplementary material.

Comments on the Quality of English Language

The quality of the images and graphs might still be improved.

Higher magnification fluorescence microscopy images can be added as supplementary material.

Author Response

Please see the rebuttal letter
